# Effects of Blackcurrant Anthocyanin on Endothelial Function and Peripheral Temperature in Young Smokers

**DOI:** 10.3390/molecules24234295

**Published:** 2019-11-25

**Authors:** Toshiko Tomisawa, Naoki Nanashima, Maiko Kitajima, Kasumi Mikami, Shizuka Takamagi, Hayato Maeda, Kayo Horie, Fu-chih Lai, Tomohiro Osanai

**Affiliations:** 1Department of Nursing Sciences, Hirosaki University Graduate School of Health Sciences, 66-1 Hon-cho, Hirosaki, Aomori 036-8564, Japan; kitajima@hirosaki-u.ac.jp (M.K.); k-mikami@hirosaki-u.ac.jp (K.M.); takamagi@hirosaki-u.ac.jp (S.T.); tosanai@hirosaki-u.ac.jp (T.O.); 2Department of Bioscience and Laboratory Medicine, Hirosaki University Graduate School of Health Sciences, 66-1 Hon-cho, Hirosaki-shi, Aomori-ken 036-8564, Japan; nnaoki@hirosaki-u.ac.jp (N.N.); k-horie@hirosaki-u.ac.jp (K.H.); 3Department of Agriculture and Life Science, Hirosaki University, 3 Bunkyo-cho, Hirosaki, Aomori 036-8561, Japan; hayatosp@hirosaki-u.ac.jp; 4College of Nursing, Taipei Medical University, 4 No. 250 Wuxing Street, Taipei 11031, Taiwan; flai@tmu.edu.tw

**Keywords:** blackcurrant, anthocyanin, smoking, peripheral temperature, endothelial function

## Abstract

Background: Blackcurrant anthocyanin (BCA) is expected to repair endothelial dysfunction, but it remains unclear whether beneficial effects are present in young healthy persons. This study examines whether supplements containing blackcurrant anthocyanin improve endothelial function and peripheral temperature in young smokers. Methods: Young, healthy male nonsmokers (N group: *n* = 11; mean age 22 ± 2 years) and smokers (S group: *n* = 13; mean age 21 ± 1 years) were enrolled. A randomized and double-blind trial was designed to compare the effects of no supplement, a supplement containing 50 mg of blackcurrant anthocyanin (supplement A), and a supplement containing 50 mg of blackcurrant anthocyanin plus vitamin E (supplement B) on flow-mediated dilatation (FMD) and skin temperature. Results: Under no supplement, FMD was unchanged during the 2 h period after smoking in the N group, whereas it was decreased during the 2 h period after smoking in the S group. Under the A supplement, FMD was decreased 1 h after smoking and returned to the baseline level 2 h after smoking in the S group. The skin temperature in the area of the foot dorsum was decreased in the S group after smoking compared with that in the N group, who did not smoke, whereas under A and B supplements, it was higher in the S group compared with that in the N group. Conclusions: BCA could attenuate the smoking-induced acute endothelial dysfunction and improve peripheral temperature in young smokers.

## 1. Introduction

Arteriosclerosis is accelerated by several risk factors, including smoking, hypertension, diabetes, and aging, and results in the occurrence of myocardial infarction or apoplexy [1,2]. Cigarette smoking is one of the most established risk factors for cardiovascular disease and the leading preventable cause of coronary artery disease and death [3]. Endothelial dysfunction is known as a disease process that occurs throughout the vascular system and is attributable to abnormal regulation of blood vessel tone and impairment of the atheroprotective properties of the normal endothelium [4,5]. It is, therefore, emerging as an important pathogenic mechanism for atherosclerosis and may be an early manifestation of cardiovascular disease [4,6]. To prevent critical cardiovascular and cerebrovascular diseases earlier, it is quite important to detect endothelial dysfunction and to improve vascular function.

With a focus on foods containing antioxidant components such as β-carotene, vitamin C, vitamin E, and polyphenol [7,8,9], flow-mediated dilatation (FMD) of the brachial artery has been widely examined in humans and improved by vitamin C and polyphenol, indicating that antioxidant components could prevent arteriosclerotic disease [10,11,12].

The blackcurrant is known as an antioxidant food rich in anthocyanin, one of the polyphenols, and has been shown to exert many profound effects in animal experiments, including inhibition of cancer cell spreading [13,14,15], hypotensive effects [16], improvement of insulin resistance [17], and antimicrobial effects [18,19]. In human research, it has also been shown to help with controlling elevated intraocular pressure in patients with glaucoma [20], improvement of retinal artery blood flow in patients with glaucoma [21], alleviation of asthenopia [22], increase in peripheral blood flow [23], improvement of lipid metabolism [24], and prevention of muscle damage [25]. Concerning arteriosclerosis, improvement of endothelial function via nitric oxide–cGMP activation was reported in hypercholesterolemic individuals [26]. These findings suggest that blackcurrant anthocyanin may inhibit the promotion of severe arteriosclerosis when given to the persons who cannot quit smoking. In the present study, we examined the effect of intake of blackcurrant anthocyanin on endothelial function and peripheral blood flow correlated with endothelial function after cigarette smoking in healthy young adults.

## 2. Results

### 2.1. FMD

As shown in Figure 1, under no supplement, FMD was unchanged in the the nonsmokers (N) group during the 2 h period of this study, whereas it was decreased from 9.5% at baseline to 8.2% 1 h after smoking one cigarette and to 8.9% 2 h after smoking one cigarette in the smokers (S) group (1 h and 2 h, both *p* < 0.05 vs. baseline). In contrast, under the A supplement, FMD was decreased from 9.5% at baseline to 8.7% 1 h after smoking one cigarette and returned to the baseline level of 9.3% 2 h after smoking one cigarette in the S group (only 1 h, *p* < 0.05 smoking vs. nonsmoking). Under the B supplement, FMD was decreased from 9.9% at baseline to 8.9% 1 h after smoking one cigarette and to 9.6% 2 h after smoking one cigarette in the S group (1 h and 2 h, both *p* < 0.05 smoking vs. nonsmoking), despite the fact that FMD in the N group was 10.4% at baseline, 10.5% after 1 h, and 10.7% after 2 h (*p* = n.s., baseline vs. 1 h and 2 h).

### 2.2. Skin Temperature

Table 1 shows changes in skin temperature under control and A and B supplements in the N and S groups. As shown in the upper panel, under no supplement, the skin temperature in the area of the left- and right-foot toes (Lt and Rt) was similar between the N and S groups during the 2 h period, although it was likely to be decreased at 2 h after smoking one cigarette in the S group. Under the A supplement, the skin temperature of Rt was similar between the N and S groups, like that under no supplement, despite the fact that in Lt, a significant difference in temperature was found between the N and S groups (*p* < 0.05). Under the B supplement, there was no difference in Lt and Rt temperatures between the N and S groups during the 2 h period.

As shown in the lower panel, under no supplement, the temperature in the area of the right foot dorsum (Rd) was decreased in the S group after smoking one cigarette, compared with that in nonsmokers (*p* < 0.05 by two way ANOVA), despite no difference in the area of the left foot dorsum (Ld) between the two groups. Under A and B supplements, the skin temperatures in the area of the left- and right-foot dorsa(Ld and Rd) were both higher in the smokers group compared with those in the nonsmokers group (all *p* < 0.05 by two-way ANOVA).

## 3. Discussion

The major findings of the present study are as follows. Under conditions with no supplement, FMD was unchanged during the 2 h period after smoking a cigarette in the N group, whereas it was decreased during this 2 h period in the S group. Under the A supplement, FMD was decreased 1 h after smoking one cigarette and returned to the baseline level 2 h after smoking one cigarette in the S group. FMD in the N group was unchanged under the A supplement at baseline through 2 h, as observed with no supplement. Under the B supplement, FMD was decreased 1 h and 2 h after smoking one cigarette in the S group. The skin temperature in the area of the foot dorsum was decreased in the S group after smoking one cigarette compared with that in the N group, whereas under both A and B supplements, it was higher in the S group compared with that in the N group.

### 3.1. Effect of Smoking on Endothelial Function

Inadequate dietary intake of antioxidant-rich foods and beverages can contribute to cardiovascular disease, and thus supplementation of dietary antioxidants has the potential to reduce oxidative stress and the risk of cardiovascular disease [27,28]. Oxidative stress can increase the risk of cardiovascular disease by causing endothelial dysfunction, which can occur well before the presentation of symptomatic cardiovascular disease [29]. Oxidative damage to endothelial cells may disturb their ability to produce nitric oxide (NO), thereby contributing to endothelial dysfunction [30]. NO is a heterodiatomic free-radical product generated through oxidation of l-arginine to l-citrulline, playing an important role in vasodilatation [31]. Its generation may be catalyzed by two different Ca^2+^/calmodulin-dependent NO synthases: the constitutively active endothelial (eNOS) and neuronal NO synthase and the Ca-insensitive inducible NO synthase (iNOS) [32]. Therefore, a balanced release of NO is associated with various important physiological functions, including relaxation of blood vessels. Anthocyanins may contribute to improving the ability of NO release, as previous studies have mentioned [33]. Vitamin E can also rescue transient impairment of endothelial function after the smoking of one cigarette [34]. Therefore, anthocyanins and Vitamin E could attenuate the transient impairment of endothelial dysfunction in smokers. However, in this study, complementing vitamin E tended to ameliorate endothelial functions in healthy nonsmokers as well as smokers, thereby blunting the beneficial effect of anthocyanin. It is important to avoid excess intake of polyphenol in healthy nonsmokers because excess polyphenol overtaxes their bodies [35]. In the present study, we showed that smoking impaired endothelial function acutely in healthy young persons, indicating the importance of prohibiting smoking in young individuals.

### 3.2. Antioxidants in Blackcurrant

Among the numerous vegetable foods analyzed, blackcurrant was included in the top list in terms of polyphenol concentration [36]. Anthocyanins are the major group of polyphenols in blackcurrant, accounting for about 80% of the total amount of quantified compounds [37,38]. Four major anthocyanins, delphinidin 3-O-glucoside, delphinidin 3-O-rutinoside, cyanidin 3-O-glucoside, and 3-O-cyanidin rutinoside, have been reported in blackcurrant [38], and delphinidin 3-O-rutinoside and 3-O-cyanidin rutinoside are specific blackcurrant anthocyanins [39]. Berry-derived anthocyanins possess high antioxidant activity [40] and neuroprotective activity in aging mice [41]. Blackcurrant also contains a wide range of flavonols, including high levels of myricetin and a relatively high amount of quercetin derivates, which possess strong neuroprotective activity [42]. Compared with other fruits, blackcurrant also has high antioxidant activity. These findings prompted us to examine the effect of blackcurrant on smoking-induced endothelial dysfunction in young smokers without obvious cardiovascular risk factors.

### 3.3. Effect of Blackcurrant on Smoking-Induced Endothelial Dysfunction

It is intriguing that a single administration of anthocyanins just prior to smoking attenuated the decrease in FMD in young subjects. Our trial with a blackcurrant supplement has added further weight to the evidence that a favorable change in endothelial function can be achieved using a controlled nutritional intervention within a healthy young population that smokes and takes no medication for cardiovascular disease risk factors.

## 4. Materials and Methods

### 4.1. Subjects

The subjects of this study were young, healthy, male nonsmokers (N group: *n* = 11, mean age 22 ± 2 years) and smokers (S group: *n* = 13, mean age 21 ± 1 years) (Table 2). The participants were free of other risk factors for coronary artery disease, and none were treated with any medications during the study. All participants provided informed consent. The subjects of the S group were smoking 14 ± 5 cigarettes a day.

### 4.2. Study Design

This study was a randomized controlled trial designed to compare the effects of two capsules containing a blackcurrant anthocyanin supplement with those of no supplement on smoking-induced acute endothelial dysfunction. The procedures of this experiment are represented in Figure 2. Subjects visited the laboratory without eating a meal or drinking coffee or tea for at least 3 h prior to testing and without taking antioxidant vitamins for the whole day. S group subjects refrained from smoking for at least 3 h before arrival in the vascular laboratory. Subjects were examined in the supine position after 15 min of rest in a quiet, air-conditioned room (22–26 °C). The setting of the measurement is shown in Figure 2. Endothelial function and peripheral blood flow were measured before and after thesubjects took the A capsule supplement, containing 50 mg of blackcurrant anthocyanin; the B capsule supplement, containing 50 mg of blackcurrant anthocyanin and vitamin E; and no capsules (i.e., no supplement). This experiment used nonprescription supplements. The compositions of the supplements are shown in Table 3. Experiments using A and B capsules were separated by a 4 day washout period. In total, the subjects made three visits over at least six days. 

### 4.3. Measurement of FMD 

Endothelium-dependent vasodilation and endothelium-independent vasodilation were measured according to methods that have been described previously. As shown in Figure 3A, brachial artery FMD was measured by a trained technician according to the guidelines for ultrasound assessment [43]. A linear-array transducer operating at 10 MHz was used to acquire longitudinal images of the right brachial artery. A standard blood pressure cuff was positioned around the right arm 5 cm below the antecubital fossa, and the artery was imaged 5 to 9 cm above the antecubital fossa. After baseline images were obtained, the cuff was inflated to 50 mmHg above systolic blood pressure for 5 min. The diastolic pre-beat diameter of the brachial artery was determined semi-automatically using an instrument equipped with software for monitoring the brachial artery diameter (Unex Co. Ltd., Nagoya, Japan); the validity and reproducibility of the equipment were confirmed elsewhere [44]. The participants rested on beds in the supine position for at least 20 min before the study.

### 4.4. Temperature of the Skin

Skin temperature was measured using a thermography device (InfRecH2640. Nippon Avionics Co.,Ltd. Tokyo, Japan) in four areas of the right- and left-foot toes (left: Lt, right: Rt) in the range below the line connecting the first and fifth metatarsal bones, and in the right- and left-foot dorsa (left: Ld, right: Rd) in the range below the line connecting the inner and external ankles, as shown in Figure 3B. The skin temperature was presented as the mean value in four areas after calculating the temperatures of 5 left- and right-foot toes and dorsa.

### 4.5. Statistical Analysis

Data were analyzed using SPSS (version 19.0) for Windows (SPSS Inc., Chicago, IL, USA). Statistical significance was defined as *p* < 0.05. For the analysis between groups before and after taking supplements, a two-way repeated-measures ANOVA was applied. The intragroup comparison between pre-intervention and postintervention was done by the Bonferroni correction for multiple comparisons.

### 4.6. Ethical Issues

All study protocols were approved by the Committee for Medical Ethics of the School of Medicine, Hirosaki University (2013-357), and informed consent was obtained from each participant prior to the study.

## 5. Conclusions

This study demonstrates that oral anthocyanins and Vitamin E supplementation can attenuate the transient impairment of smoking-induced acute endothelial dysfunction and peripheral blood flow in smokers. These data are consistent with the idea that polyphenols, including anthocyanin, may in part be beneficial, due to direct tissue effects, in the prevention of cardiovascular diseases.

## Figures and Tables

**Figure 1 molecules-24-04295-f001:**
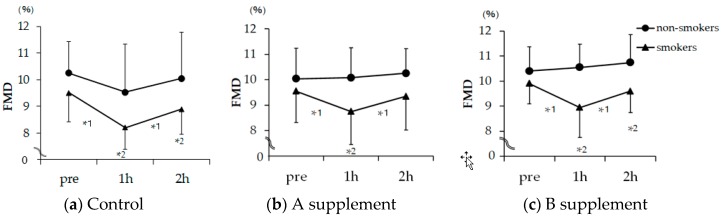
Changes in brachial artery flow-mediated dilation (FMD) in smokers and non-smokers receiving no supplement (Control) (**a**), A supplement (**b**), and B supplement (**c**) measured before and after moking one cigarette. Circular mark: nonsmokers; triangular mark: smokers. FMD is shown by the percentage change of the diameter of the brachial artery at maximal dilatation during reactive hyperemia compared to the baseline value. Values are indicated as means ± standard errors. Difference in changes from baseline to 2 h between groups was assessed by two-way repeated measure ANOVA followed by Bonferroni adjustment as a post-hoc test. The B supplement induced a statistically significant difference by two-way repeated measure ANOVA (*p* < 0.05), but there were no statistically significant differences between the control and the A supplement group. *1 was analyzed by Bonferroni test between times on each group, and *2 was analyzed by Bonferroni adjustment between groups; *1 and *2 mean statistical significance was *p* < 0.05.

**Figure 2 molecules-24-04295-f002:**
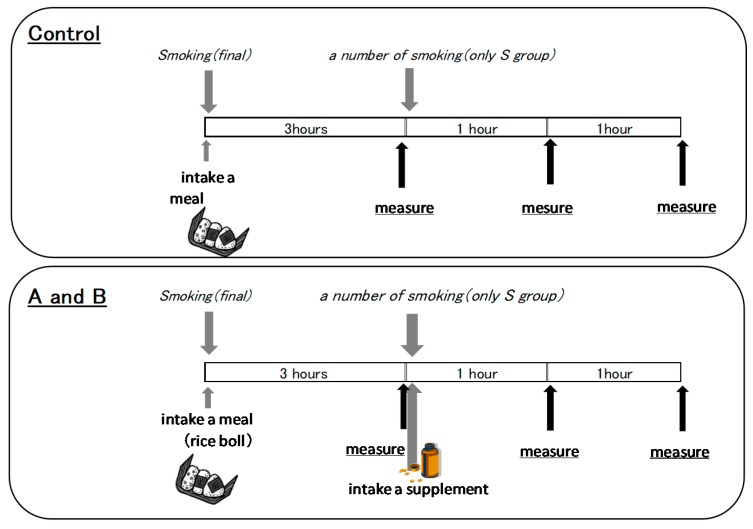
Study protocol.

**Figure 3 molecules-24-04295-f003:**
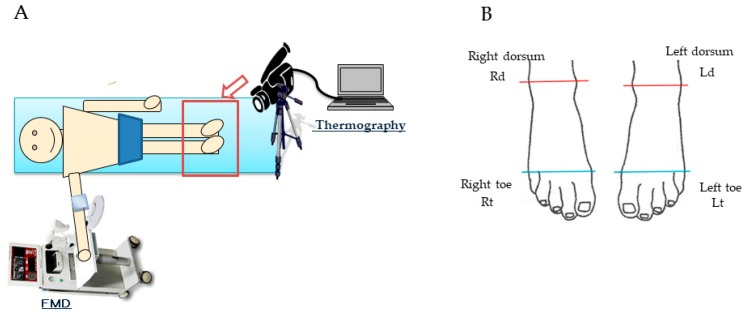
Setting and Measurement site. (**A**) shows the setting during measuring outcome. (**B**) shows 4 temparature-measued areas of the right- and left-foot toes (left: Lt, right: Rt) in the range below the line connecting the first and fifth metatarsal bones, and in the right- and left-foot dorsa (left: Ld, right: Rd) in the range below the line connecting the inner and external ankles.

**Table 1 molecules-24-04295-t001:** Changes in skin temperature.

		N Group (*n* = 11)			S Group (*n* = 13)			2-Way Repeated Measure ANOVA
		pre	post1	post2	pre	post1	post2	
Lt	Control	29.2	29.5	28.6	29.7	28.8	27.7	n.s.
	A	29.4	28.9	28.3	28.4	28.7	30.0	*p* < 0.05
	B	27.6	26.5	26.2	28.6	27.6	28.4	n.s.
Rt	Control	29.1	29.3	28.6	29.8	28.8	26.9	n.s.
	A	29.1	28.8	28.0	28.2	28.8	29.5	n.s.
	B	27.5	26.6	26.4	28.5	27.3	28.1	n.s.
Ld	Control	30.5	30.6	30.5	31.2	30.1	29.9	n.s.
	A	31.2	30.7	29.3	30.1	30.3	31.2	*p* < 0.01
	B	30.1	29.6	29.0	30.2	29.5	30.2	*p* < 0.05
Rd	Control	30.3	30.6	30.7	31.4	30.3	29.5	*p* < 0.05
	A	31.0	30.4	30.1	30.1	30.4	30.8	*p* < 0.05
	B	30.3	29.5	29.2	30.1	29.6	30.2	*p* < 0.05

Changes in skin temperature with nosupplement (control) (**a**), A supplement (**b**), and B supplement (**c**) between smokers and nonsmokers before and after smoking a cigarette. Values are indicated as the means of skin temperature (°C). Lt and Rt mean left- and right-foot toes. Ld and Rd represent left- and right-foot dorsum. Difference in change for each supplement between smokers and nonsmokers was assessed by two-way repeated measure ANOVA; n.s. represents no statistically significant change. *p* < 0.05 means statistically significant change between smokers and nonsmokers.

**Table 2 molecules-24-04295-t002:** Characteristics of the enrolled subjects. BMI: body mass index.

		Non-Smoking Group (*n* = 13)	Smoking Group (*n* = 11)
age (year)		22 ± 2.1	21 ± 0.9
BMI (kg/m^2^)		20.9 ± 1.4	21.3 ± 2.1
a number of cigarette (number)		0	14 ± 5
nicotine (mg/a cigarette)		0	1.5 ± 2.0
tar (mg/a cigarette)		0	10.9 ± 3.9
Food Habit	1 meal	0	1
	2 meals	4	2
	3 meals	5	7
	much difference	2	3
caffeine intake	every day	2	5
	sometimes	3	6
	no intake	6	2
Sleeping time	less than 3 h	1	0
	4–6 h	4	2
	6–8 h	5	6

Table 2 shows characteristics of the enrolled subjects. There were no differences in characteristics and daily habits between the two groups.

**Table 3 molecules-24-04295-t003:** Nutrition composition of A and B supplement capsules.

	A capsule	B capsule
weight (g)	1.02	1.47
calorie (kcal)	5.7	7.9
protein (g)	0.32	0.47
lipid (g)	0.36	0.49
carbohydrate (g)	0.3	0.4
Na (mg)	<5	9.5
βcarotene (μg)	1800	2250
Vitamine C (mg)		50
Vitamine E (mg)		27
Cu (mg)		0.6
Zn (mg)		9
BCA (mg)	50	50
lutein (mg)	0.5	12

Table 3 shows composition of the supplements. BCA means blackcurrant anthocyanin.

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
