# Peer review of "Effects of Blackcurrant Anthocyanin on Endothelial Function and Peripheral Temperature in Young Smokers"

_molecules, 2019, doi:10.3390/molecules24234295_

Round 1
Reviewer 1 Report
Effects of blackcurrant anthocyanin on endothelial function and peripheral temperature in young smokers
The authors in the current study investigated the effect of Blackcurrant Anthocyanin (BCA) on endothelial dysfunction. They reported that BCA administration could attenuate the smoking-induced acute endothelial dysfunction and improve peripheral temperature in young smokers. The introduction section of the manuscript should be improved. The rationale of measuring the skin temperature should be clearly explained and why is this related to endothelial dysfunction.
Please find below specific comments.
1, the authors are comparing the FMD results between smokers and non smokers. What about the FMD changes from baseline to 1h then 2h, were those changes significant overtime in each group? 1, what does the authors mean by the (%) in FMD y-axis? Percent of what? This should be explained clearly in the figure legend. Line 77, “ANOVA(*1) followed by Bonferroni post hoc tests(*2). *1 and *2 means statistical significance as p < 0.05. “ is very confusing and I do not understand what they mean by those comparisons. The authors should clarify this in the figure legends. The authors have to clearly explain why they measured the skin temperature and what this has to do with the purpose of the study. The authors should comment on why the right foot temperature decreased but not the left foot. Table 1, there is no table legend. Authors should also explain in the legend P<0.05 is significant versus which groups. Table 2 does not have a table legend. Line 174, Do the authors mean by “4-day washout period” that they gave the 2 supplements to the same subject? This should be explained in the methods section clearly. If so, why the authors did not use different subjects for each supplement to exclude mixed effects and why they chose 4-days washout period? Is there any pharmacokinetics studies done on those capsules? Line 180, a reference is missing. Table 3 does not have a table legend. Table 3, where is the anthocyanin in the capsules composition? Line 207, a reference is missing. If the anthocyanin is beneficial in reducing endothelial dysfunction, what could be the potential mechanisms and did any of those potential proteins change in the current study with treatment?
Author Response
Dear Reviewer1
Thank you for the opportunity to revise our manuscript. I uploaded the letter for you. Please see the attachment.

Reviewer 2 Report
The manuscript entitled "Effects of blackcurrant anthocyanin on endothelial function and peripheral temperature in young smokers" is an interesting study that will contribute to the current literature. I recommend the publication of this paper in “Molecules” after the minor amendments given below:
- Lines 40-42: “It is, therefore, emerging as an important pathogenic mechanism for atherosclerosis and may be an early manifestation of cardiovascular disease [4,5]”. Here I suggest authors to also refer to the following publication: doi: 10.3390/nu7115462.
- Line 73: “This” should be revised as “The”.
- Table 1: Please indicate the abbreviations presented in this table in full as a footnote.
- Line 180 and 207: Please indicate the related references.
- Table 3: Please delete the footnote.
- It might be good to write a brief conclusion section.
Author Response
Dear Reviewer2,
Thank you for the opportunity to revise our manuscript. I uploaded the cover letter for you. Please see the attachment.

Round 2
Reviewer 1 Report
Please find below the comments that were not clearly addressed in the revised version.
comment 1: a reference should be included to support the authors' rationale. Comment 2: the legend provided by the authors is still confusing and my comment were not addressed completely. The percent is not explained. Also, why the authors chose to use Student t-test as a post-hoc test versus using Bonferroni? They should stick to one post-hoc test. Comment 4: P<0.05 is still not explained it is significant versus which groups. Do the authors mean it is a significant change over time? Comment 6: the authors did not address the rest of the question. This is should be explained and justified clearly in the manuscript. Comment 10: the authors wrote "There are several evidences to improve endothelial dysfunction by anthocyanins." What are those several evidences mentioned in the text that anthocyanins can improve endothelial dysfunction? the authors explained the importance of NO for endothelial dysfunction without really commenting on how the anthocyanin can affect this.
Author Response
Dear Reviewer 1,
Thank you for your great and kind suggestion. Please see the attachement.
